# THREE CONTINUAL LEARNING SCENARIOS AND A CASE FOR GENERATIVE REPLAY

## ABSTRACT

Standard artificial neural networks suffer from the well-known issue of catastrophic forgetting, making continual or lifelong learning problematic. Recently, numerous methods have been proposed for continual learning, but due to differences in evaluation protocols it is difficult to directly compare their performance. To enable more meaningful comparisons, we identified three distinct continual learning scenarios based on whether task identity is known and, if it is not, whether it needs to be inferred. Performing the split and permuted MNIST task protocols according to each of these scenarios, we found that regularization-based approaches (e.g., elastic weight consolidation) failed when task identity needed to be inferred. In contrast, generative replay combined with distillation (i.e., using class probabilities as "soft targets") achieved superior performance in all three scenarios. In addition, we reduced the computational cost of generative replay by integrating the generative model into the main model.

## 1 INTRODUCTION

Current state-of-the-art deep neural networks can be trained to impressive performance on a wide variety of individual tasks. Learning multiple tasks in sequence, however, remains a substantial challenge for deep learning. When trained on a new task, standard neural networks forget most information related to previously learned tasks, a phenomenon referred to as "catastrophic forgetting".

In recent years, numerous methods for alleviating catastrophic forgetting have been proposed. However, due to the wide variety of experimental protocols used to evaluate them, many of these methods claim "state-of-the-art" performance (e.g., Kirkpatrick et al., 2017; Rebuffi et al., 2017; Nguyen et al., 2017; Masse et al., 2018; Kemker & Kanan, 2018; Wu et al., 2018). To obscure things further, some methods shown to perform well in some experimental settings are reported to dramatically fail in others: compare the performance of elastic weight consolidation in Kirkpatrick et al. (2017) and Zenke et al. (2017) with that in Kemker et al. (2017) and Kamra et al. (2017).

To enable a fairer and more structured comparison of methods for reducing catastrophic forgetting, as a first contribution this paper identifies three distinct continual learning scenarios of increasing difficulty. These scenarios are distinguished by whether at test time task identity is provided and, if it is not, whether task identity needs to be inferred. We show that such differences in experimental design can explain seemingly contradictory results reported in the recent literature: even for experimental protocols involving the relatively simple classification of MNIST-digits, methods that perform well in one continual learning scenario can completely fail in another.

Using these three scenarios, a second contribution of this paper is to provide an extensive comparison of recently proposed methods. These experiments reveal that generative replay, especially when combined with distillation techniques, has the capability to perform well on all three scenarios. An important disadvantage of this approach, however, is that it can be computationally very costly.

As a third contribution, this paper proposes a way to reduce these computational costs. Current approaches using generative replay train two separate models: a main model for solving the tasks and a generative model for sampling examples representative of previous tasks. We merge the generative model into the main model by equipping it with backward connections that are trained to have generative capability (e.g., a variational autoencoder with added softmax layer). We demonstrate that this can substantially reduce training time, with no or negligible loss in performance.

## 2 CONTINUAL LEARNING SCENARIOS

We consider the continual learning problem in which a single model needs to sequentially learn a series of tasks, whereby it is not allowed to store raw data. This continual learning framework has been actively studied in recent years: many methods for alleviating catastrophic forgetting are being proposed, with almost as many different experimental protocols being used for their evaluation. We found that an important difference between these experimental protocols is whether at test time information about the task identity is available and—if it is not—whether the model is required to identify the identity of the task it has to solve. Yet, this crucial experimental design consideration is not always clearly stated and differences in this regard are sometimes not appreciated. For example, in Masse et al. (2018) a substantial improvement over state-of-the-art is reported, while their method assumes task identity is always available and the compared methods operate without this assumption. To enable more meaningful comparisons, we identify three distinct scenarios for continual learning.

In the first scenario, models are always informed about which task needs to be performed. This is the easiest continual learning scenario, and we refer to it as **incremental task learning**. Since task identity is always provided, it is possible to train models with task-specific components. A typical neural network architecture used in this scenario has a "multihead" output layer, meaning that each task has its own output units but the rest of the network is (potentially) shared between tasks.

In the second scenario, which we refer to as **incremental domain learning**, task identity is not available at test time. Models however only need to solve the task at hand; they are not required to infer which task it is. Typical examples of this scenario are protocols whereby the structure of the tasks is always the same, but the input-distribution is changing. A classical example of such a task protocol is 'permuted MNIST' (Goodfellow et al., 2013), in which all tasks involve classifying MNIST-digits but with a different permutation applied to the pixels for each new task (Figure 2). Although permuted MNIST is most naturally performed according to the incremental domain learning scenario, it can be performed according to the other scenarios too (Table 2).

Finally, in the third scenario, models need to be able both to solve each task seen so far and to infer which task they are presented with. We refer to this scenario as **incremental class learning**, as it includes protocols in which new classes need to be learned incrementally. An example task protocol most naturally performed under this scenario is sequentially learning MNIST-digits ('split MNIST'; Figure 1), although this protocol has also been performed under the other two scenarios (Table 1).

## 3 CONTINUAL LEARNING STRATEGIES

A simple and intuitive explanation for catastrophic forgetting is that after a neural network is trained on a new task, its parameters are now optimized for the new task and no longer for the previous one(s). This formulation highlights two strategies for alleviating catastrophic forgetting: (1) not freely optimizing the entire network on each task, and (2) modifying the training data to make it more representative for previous tasks.

### 3.1 NOT OPTIMIZING ENTIRE NETWORK / REGULARIZED OPTIMIZATION

A straightforward way of not optimizing the full network on every task is to explicitly define a different subnetwork per task. Several recent papers use this strategy, with different approaches for selecting the parts of the network for each task. A simple approach is to randomly assign which nodes participate in each task (*Context-dependent Gating* [XdG]; Masse et al., 2018). Other approaches use evolutionary algorithms (Fernando et al., 2017) or gradient descent (Serrà et al., 2018) to learn which units to employ for each task. By design, these approaches are limited to the incremental task learning scenario, as task identity is required to select the correct task-specific components.

A modification to make this strategy applicable in the other scenarios is to preferentially train a different part of the network for each task, but to always use the entire network for execution. One way to do this is by differently regularizing the network's parameters during training on each new task, which is the approach of *Elastic Weight Consolidation* (EWC; Kirkpatrick et al., 2017) and *Synaptic Intelligence* (SI; Zenke et al., 2017). Both methods estimate for all parameters of the network how important they are for the previously learned tasks and penalize future changes to them accordingly (i.e., learning is slowed down for parts of the network important for previous tasks).

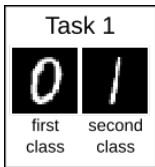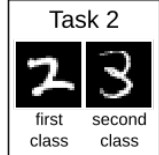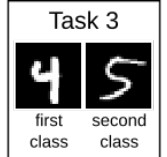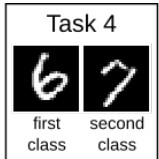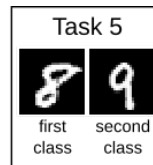

Figure 1: Schematic of the split MNIST task protocol.

Table 1: The split MNIST task protocol according to each continual learning scenario.

| | |
|---|---|
| **Incremental task learning** | With task given, is it the first or second class? (e.g., '0' or '1') |
| **Incremental domain learning** | With task unknown, is it a first or second class? (e.g., in ['0','2','4','6','8'] or in ['1','3','5','7','9']) |
| **Incremental class learning** | With task unknown, which digit is it? (choice from '0' to '9') |

## 3.2 MODIFYING TRAINING DATA

A second strategy is to complement the training data for each new task to be learned with "pseudo-data" representative of the previous tasks. We refer to this strategy as replay. An early implementation of this strategy, called pseudo-rehearsal, generated completely random inputs as pseudo-data and labeled them based on the predictions of a copy of the model stored after finishing training on the previous task (Robins, 1995). This approach had some success with very simple, artificial inputs, but does not work with more complicated inputs (Atkinson et al., 2018).

An alternative is to take the input data of the current task, label them using the model trained on the previous tasks, and use the resulting input-target pairs as pseudo-data. This is the approach of *Learning without Forgetting* (LwF; Li & Hoiem, 2017). Another important aspect of this method is that instead of labeling the inputs to be replayed as the most likely category according to the previous tasks' model (i.e., "hard targets"), it pairs them with the by that model predicted probabilities for *all* target classes (i.e., "soft targets"). The objective for the replayed data is then to match the probabilities predicted by the model being trained to these target probabilities. The approach of matching predicted (and typically temperature-raised) probabilities of one network to those of another network had previously been used to compress (or "distill") information from one (large) network to another (smaller) network (Hinton et al., 2015).

Another option is to generate the input data to be replayed. For this, besides the main model for task performance (e.g., classification), a separate generative model is sequentially trained on all tasks to generate samples from their input data distributions. For the first application of this approach, which was called *Deep Generative Replay* (DGR), the generated input samples were paired with "hard targets" provided by the main model (Shin et al., 2017). We note that it is possible to combine LwF and DGR by replaying input samples from a generative model and pairing them with soft targets (see also Wu et al., 2018; Venkatesan et al., 2017). We include this hybrid method in our comparison under the name DGR+distill.

A final option is to store examples from previous tasks and replay those. Such "exact replay" can substantially boost performance (Rebuffi et al., 2017; Nguyen et al., 2017; Kemker & Kanan, 2018; see Appendix C), but due to privacy concerns or memory constraints, it is not always possible to do so. In this paper we restrict ourselves to the case where storing raw data is not allowed.

## 4 EXPERIMENTAL DETAILS

To compare the performance of the above discussed approaches, we used two different task protocols that were both performed according to all three continual learning scenarios defined in section 2.

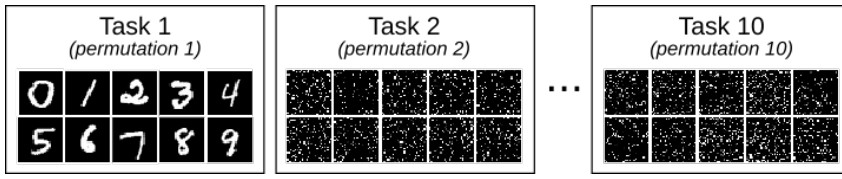

Figure 2: Schematic of the permuted MNIST task protocol.

Table 2: The permuted MNIST task protocol according to each continual learning scenario.

| | |
|---|---|
| **Incremental task learning** | Given permutation $X$ was applied, which digit is it? |
| **Incremental domain learning** | With permutation unknown, which digit is it? |
| **Incremental class learning** | Which digit is it *and* which permutation was applied? |

## 4.1 TASK PROTOCOLS

The first task protocol was split MNIST (Zenke et al., 2017; Figure 1). For this, the original MNIST-dataset was split into five tasks, where each task was a two digit classification. The original 28x28 pixel grey-scale images were used without pre-processing. The standard training/test-split was used resulting in 60,000 training images (~6000 per digit) and 10,000 test images (~1000 per digit).

The second task protocol was permuted MNIST (Goodfellow et al., 2013; Figure 2). The tasks of this protocol were classifying MNIST-digits (every task now had all ten digits), whereby in each task the pixels of the MNIST-images were permutated in a different way. We used a sequence of ten such tasks. To generate the permutated images, the original images were first zero-padded to 32x32 pixels. For each task, a random permutation was then generated and applied to these 1024 pixels. No other pre-processing was performed. Again the standard training/test-split was used.

## 4.2 METHODS

For a fair comparison, the same neural network architecture was used for all methods. This was a multi-layer perceptron with 2 hidden layers of 400 (split MNIST) or 1000 (permuted MNIST) nodes each, followed by a softmax output layer. ReLU non-linearities were used in all hidden layers. In the incremental task learning scenario, all methods used a "multihead" output layer, meaning that each task had its own output units and always only the output units of the task under consideration—i.e., either the current task or the replayed task—were active. In the incremental domain learning scenario, all methods were implemented with a "singlehead" output layer, meaning that each task used the same output units (with each unit corresponding to one class in every task). In the incremental class learning scenario, each class had its own output unit and always all units of the classes seen so far were active (see also Appendices A.1.1 and A.1.2).

All methods used the standard cross entropy classification loss for the model's predictions on the current task data ($\mathcal{L}_{\text{current}} = \mathcal{L}_{\text{classification}}$; see Appendix A.1.1). The regularization-based methods (i.e., EWC, online EWC and SI) added a regularization term to this loss, with regularization strength controlled by a hyperparameter: $\mathcal{L}_{\text{total}} = \mathcal{L}_{\text{current}} + \lambda \mathcal{L}_{\text{regularization}}$. The value of this hyperparameter was set by a grid search, even though it could be argued that this is problematic in the context of continual learning (see Appendix B). The replay-based methods (i.e., LwF, DGR and DGR+distill) instead added a loss-term for the replayed data. In this case a hyperparameter could be avoided, as the loss for the current and replayed data could be weighted according to how many tasks the model has been trained on so far: $\mathcal{L}_{\text{total}} = \frac{1}{N_{\text{tasks so far}}} \mathcal{L}_{\text{current}} + \left(1 - \frac{1}{N_{\text{tasks so far}}}\right) \mathcal{L}_{\text{replay}}$.

We compared the following approaches:

- **None:** The model was sequentially trained on all tasks in the standard way. This is also called *fine-tuning*, and can be seen as a lower bound.

- **XdG:** Following Masse et al. (2018), for each task a random subset of $X\%$ of the units in each hidden layer was fully gated (i.e., their activations set to zero), with $X$ a hyperparameter whose value was set by a grid search (see Appendix B). As this method requires availability of task identity at test time, it could only be used in the incremental task learning scenario.

- **EWC:** The regularization term proposed in Kirkpatrick et al. (2017) was added to the loss, see Appendix A.2.1 for implementation details.

- **Online EWC:** This is a modification of EWC proposed by Schwarz et al. (2018), with inspiration from Huszár (2018), that improves EWC's scalability by ensuring the computational cost of the regularization term does not grow with number of tasks (see Appendix A.2.2).

- **SI:** The regularization proposed in Zenke et al. (2017) was added to the loss (see Appendix A.2.3).

- **LwF:** Images of the current task were replayed with soft targets provided by a copy of the model stored after finishing training on the previous task (Li & Hoiem, 2017; see Appendix A.1.2).

- **DGR:** A separate generative model was trained to generate the images to be replayed. Following Shin et al. (2017), the replayed images were labeled with the most likely category predicted by a copy of the main model stored after training on the previous task (i.e., hard targets).

- **DGR+distill:** A separate generative model was trained to generate the images to be replayed, but these were then paired with soft targets (as in LwF) instead of hard targets (as in DGR).

- **Offline:** The model was always trained using the data of all tasks so far. This is also called *joint training*, and was included as it can be seen as an upper bound.

For the split MNIST protocol, all models were trained for 2000 iterations per task using the ADAM-optimizer ($\beta_1 = 0.9$, $\beta_2 = 0.999$; Kingma & Ba, 2014) with learning rate 0.001. The same optimizer was used for the permuted MNIST protocol, but with 5000 iterations and learning rate 0.0001. For each iteration, $\mathcal{L}_{\text{current}}$ (and $\mathcal{L}_{\text{regularization}}$) was calculated as average over 128 examples from the current task and—if replay was used—an additional 128 replayed examples (equally divided over all previous tasks) were used to calculate $\mathcal{L}_{\text{replay}}$. Importantly, since the total number of replayed examples does not depend on the number of previous tasks, for our implementation of the replay-based methods the training time per task does not need to increase with number of tasks so far.

For DGR and DGR+distill, a separate generative model was sequentially trained on all tasks. A symmetric variational autoencoder (VAE; Kingma & Welling, 2013) was used as generative model, with 2 fully connected hidden layers of 400 (split MNIST) or 1000 (permuted MNIST) units and a stochastic latent variable layer of size 100. A standard normal distribution was used as prior. See Appendix A.1.3 for more details. Training of the generative model was also done with generative replay (provided by its own copy stored after finishing training on the previous task) and with the same hyperparameters (i.e., learning rate, optimizer, iterations, batch sizes) as for the main model.

## 5 RESULTS

For the split MNIST task protocol, we found a clear difference in difficulty between the three continual learning scenarios (see Table 3). Perhaps surprisingly, for all three scenarios with the split MNIST protocol, EWC and online EWC barely outperformed fine-tuning. SI performed better: it reduced catastrophic forgetting in the incremental task learning and incremental domain learning scenarios, but it also failed in the incremental class learning scenario. Strikingly, replaying images from the current task (LwF; e.g., replaying '2's and '3's in order not to forget how to recognize '0's and '1's), prevented the forgetting of previous tasks better than SI. Importantly, only the methods using generative replay retained good performance (above 90%) in the incremental class learning scenario, and DGR+distill outperformed DGR in all scenarios.

For the permuted MNIST protocol (see Table 4), there was less difference between EWC, online EWC and SI: they performed reasonably well in the incremental task learning and incremental domain learning scenarios, but failed again in the incremental class learning scenario. While LwF had some success with the split MNIST protocol, this method did not work with the permuted MNIST protocol. The methods using generative replay were again the only ones successful in the incremental class learning scenario, and DGR+distill again always outperformed DGR.

Finally, although in the incremental task learning scenario XdG succeeded in reducing catastrophic forgetting on both task protocols, on both it was outperformed by SI (and thus by DGR+distill).

Table 3: Average test accuracy (over all tasks) on the split MNIST task protocol. Each experiment was performed 20 times with different random seeds, reported is the mean ($\pm$ SEM) over these runs.

| Method | Incremental task learning | Incremental domain learning | Incremental class learning |
|---|---|---|---|
| None – *lower bound* | 85.15 ($\pm$ 1.00) | 57.33 ($\pm$ 1.66) | 19.90 ($\pm$ 0.02) |
| XdG | 98.74 ($\pm$ 0.31) | - | - |
| EWC | 85.48 ($\pm$ 1.20) | 57.80 ($\pm$ 1.61) | 19.90 ($\pm$ 0.02) |
| Online EWC | 85.22 ($\pm$ 1.06) | 57.60 ($\pm$ 1.66) | 19.90 ($\pm$ 0.02) |
| SI | 99.14 ($\pm$ 0.11) | 63.77 ($\pm$ 1.18) | 20.04 ($\pm$ 0.08) |
| LwF | 99.60 ($\pm$ 0.03) | 71.02 ($\pm$ 1.26) | 24.17 ($\pm$ 0.51) |
| DGR | 99.47 ($\pm$ 0.03) | 95.74 ($\pm$ 0.23) | 91.24 ($\pm$ 0.33) |
| DGR+distill | 99.59 ($\pm$ 0.03) | 96.94 ($\pm$ 0.14) | 91.84 ($\pm$ 0.27) |
| IGR (see below) | 99.66 ($\pm$ 0.03) | 97.31 ($\pm$ 0.11) | 92.56 ($\pm$ 0.21) |
| Offline – *upper bound* | 99.64 ($\pm$ 0.03) | 98.41 ($\pm$ 0.06) | 97.93 ($\pm$ 0.04) |

Table 4: Idem as Table 3, except on the permuted MNIST task protocol.

| Method | Incremental task learning | Incremental domain learning | Incremental class learning |
|---|---|---|---|
| None – *lower bound* | 81.79 ($\pm$ 0.48) | 78.51 ($\pm$ 0.24) | 17.26 ($\pm$ 0.19) |
| XdG | 91.40 ($\pm$ 0.23) | - | - |
| EWC | 94.74 ($\pm$ 0.05) | 94.31 ($\pm$ 0.11) | 25.04 ($\pm$ 0.50) |
| Online EWC | 95.96 ($\pm$ 0.06) | 94.42 ($\pm$ 0.13) | 33.88 ($\pm$ 0.49) |
| SI | 94.75 ($\pm$ 0.14) | 95.33 ($\pm$ 0.11) | 29.31 ($\pm$ 0.62) |
| LwF | 69.84 ($\pm$ 0.46) | 72.64 ($\pm$ 0.52) | 22.64 ($\pm$ 0.23) |
| DGR | 92.52 ($\pm$ 0.08) | 95.09 ($\pm$ 0.04) | 92.19 ($\pm$ 0.09) |
| DGR+distill | 97.51 ($\pm$ 0.01) | 97.35 ($\pm$ 0.02) | 96.38 ($\pm$ 0.03) |
| IGR (see below) | 97.31 ($\pm$ 0.01) | 97.06 ($\pm$ 0.02) | 96.23 ($\pm$ 0.04) |
| Offline – *upper bound* | 97.68 ($\pm$ 0.01) | 97.59 ($\pm$ 0.01) | 97.59 ($\pm$ 0.02) |

# 6 INTEGRATED GENERATIVE REPLAY (IGR)

Generative replay with distillation consistently outperformed the competing methods and even obtained excellent results in the challenging incremental class learning scenario. However, an important disadvantage of generative replay is that it is usually computationally expensive, among others because a separate generative model is trained. Indeed, in our experiments the training time for DGR and DGR+distill was roughly twice as long as for SI (see below). To reduce the computational cost of generative replay, we propose to integrate the generative model into the main model by equipping it with generative backward connections.

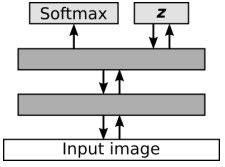

Figure 3: IGR schematic.

**Theory**   To enable the main model to generate replay itself, we add (1) backward connections that are trained to reconstruct inputs from their hidden representations and (2) a layer of stochastic latent variables $z$ that are trained to follow a known distribution from which it is easy to sample. In case of classification, the resulting network is for example a symmetrical VAE with an additional softmax classification layer from the final hidden layer of the encoder (Figure 3). Besides removing the need for a separate generative model, it is possible that regularization provided by the added generative objective helps to train a more robust classifier (Lasserre et al., 2006; Kingma et al., 2014).

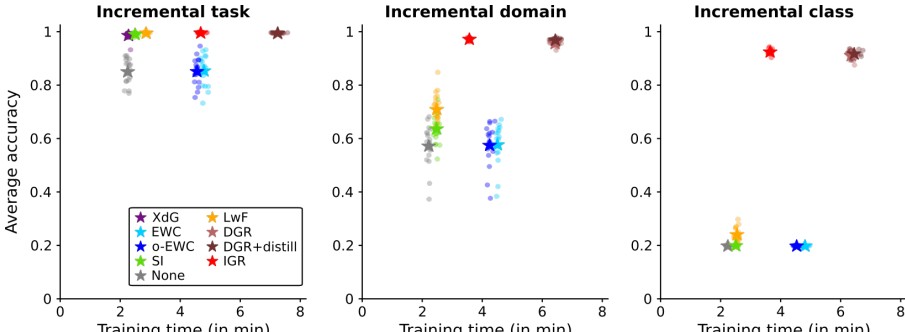

Figure 4: Average test accuracy (over all tasks) on the split MNIST protocol plotted against training time. Each experiment was run 20 times: dots represent individual runs, stars indicate the mean.

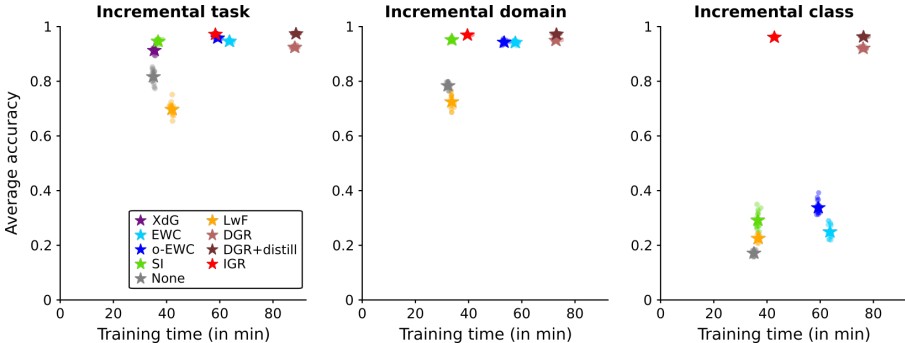

Figure 5: Idem as Figure 4, except on the permuted MNIST task protocol.

The loss function for the data of the current task now has two terms: $\mathcal{L}_{\text{current}} = \mathcal{L}_{\text{generative}} + \mathcal{L}_{\text{classification}}$, whereby $\mathcal{L}_{\text{classification}}$ is the standard cross-entropy classification loss and $\mathcal{L}_{\text{generative}}$ is the VAE loss (see Appendix A.1.3). On later tasks, the training data of the current task is supplemented with replayed data. For the replayed data, as for LwF and DGR+distill, the classification term is replaced by a distillation term: $\mathcal{L}_{\text{replay}} = \mathcal{L}_{\text{generative}} + \mathcal{L}_{\text{distillation}}$. The loss terms for the current and replayed data are again weighted according to how many tasks the model has seen so far.

**Results**   For a fair comparison, the model used for IGR also had 2 fully connected hidden layers with 400 (split MNIST) or 1000 (permuted MNIST) units. Similar to the VAE used in DGR and DGR+distill, the stochastic latent variable layer was of size 100 with a standard normal prior. Also the same hyperparameters (i.e., learning rate, optimizer, iterations, batch size) were used for training.

We found that IGR slightly outperformed DGR+distill on all experiments with the split MNIST protocol (Table 3), while it performed slightly less on the experiments with the permuted MNIST protocol (Table 4). These differences were relatively small and, similar to DGR+distill, IGR comfortably outperformed all other tested methods. To assess the extent to which IGR reduced the computational cost of generative replay, and to compare the resulting cost with that of the other methods, in Figures 4 and 5 we plotted for each method its performance against total training time on a NVIDIA GeForce GTX 1080 GPU. As expected, training time was longest for DGR and DGR+distill, and this time was substantially reduced—for most experiments almost halved—by IGR.

## 7   DISCUSSION

Catastrophic forgetting is a major obstacle to the development of artificial intelligence applications capable of true lifelong learning (Kumaran et al., 2016; Parisi et al., 2018), and enabling neural networks to sequentially learn multiple tasks has become a topic of intense research. Despite its

After task 1       After task 4       After task 6       After task 9

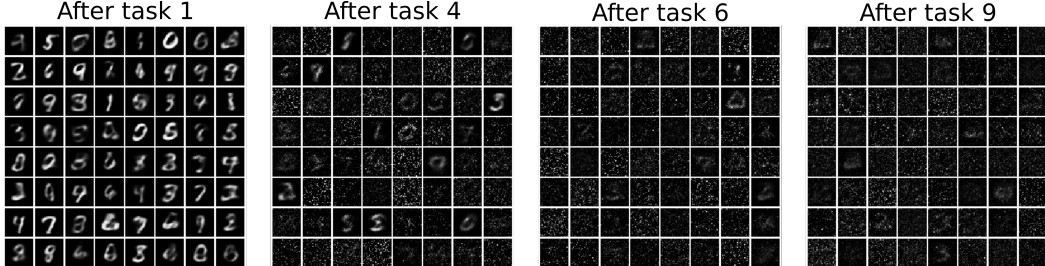

Figure 6: Randomly selected samples from the generative model after finishing training on task 1, 4, 6 and 9 of the permuted MNIST protocol (i.e., examples of what is replayed during task 2, 5, 7 and 10). No permutation was applied for task 1. Because after task 6 (or 9), ~17% (or ~11%) of samples should correspond to task 1 and should thus be unshuffled digits, this figure shows that at least the quality of those replayed images is far from perfect. Nevertheless, task 1 is not being forgotten.

scope, this research field lacks common benchmarks—even though the same datasets tend to be used, which makes direct comparisons between published methods difficult. We found that an important difference between currently used experimental protocols is in whether task identity is provided and—if it is not—in whether it needs to be inferred. For each of the resulting three scenarios, we performed a comprehensive comparison of recently proposed methods. An important conclusion is that for the incremental class learning scenario (i.e., when task identity needs to be inferred), only replay-based methods are capable of producing acceptable results. In this ethological most relevant scenario, even for relatively simple task protocols involving the classification of MNIST-digits, regularization-based methods such as EWC and SI completely failed. Moreover, also in the other scenarios, generative replay combined with distillation consistently outperformed all other tested methods. These results establish generative replay as a promising general strategy for lifelong learning.

However, an important limitation of the current study is that generating MNIST-digits is relatively easy. We leave it for future work to empirically address whether generative replay can scale to task protocols with more complicated inputs, but here we highlight several reasons why we believe this will be the case. First, with the permuted MNIST protocol, we observed that even when the quality of the replayed samples had substantially declined (see Figure 6), they still helped to prevent catastrophic forgetting. Second, under some conditions (e.g., with the split MNIST protocol), replaying inputs from the current task (i.e., LwF) works reasonably well, further indicating that the replayed samples need not be perfect and that "good enough" can suffice. We hypothesize that the use of distillation is especially important to make generative replay more robust to the quality of the replayed inputs. Finally, of course, the capabilities of generative models are improving at a rapid pace (e.g., Goodfellow et al., 2014; Oord et al., 2016; Rezende & Mohamed, 2015).

This last point however also warrants caution. Although the latest developments in for example generative adversarial networks, auto-regressive decoders or flow-based models enable training high quality generative models for increasingly complicated input distributions, this can come at high computational costs. Especially in a lifelong learning setting, where models continually need to be trained on new tasks and where training sometimes has to be in real-time, efficiency is important. We therefore emphasize that continual learning methods should not only be evaluated in terms of their performance, but also in terms of for example their training time (e.g., Figures 4 and 5; see also Farquhar & Gal, 2018). Here, we improved the efficiency of generative replay by merging the generator into the main model. We also want to highlight that in our implementation of replay the number of replayed examples did not increase with number of tasks. We hypothesize that a relatively small number of examples per task can be acceptable because information on the previous tasks is also contained in the initiation bias (i.e., training on each new task starts with a network that is already optimized for the previous tasks).

To conclude, we believe that generative replay brings more to the table than simply "*shift[ing] the catastrophic forgetting problem to the training of the generative model*" (Schwarz et al., 2018; p.3), and we envision that a small amount of good enough replay generated by the model's own backward connections could become a valuable tool for real-world continual learning applications.

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

# A  ADDITIONAL EXPERIMENTAL DETAILS

PyTorch-code that can be used to perform the experiments described in this paper is available online: `link-to-appear-here-in-deanonymized-version`.

## A.1  LOSS FUNCTIONS

### A.1.1  CLASSIFICATION

The standard per-sample cross entropy loss function for an input $\boldsymbol{x}$ labeled with a hard target $y$ is given by:

$$\mathcal{L}_{\text{classification}}\left(\boldsymbol{x}, y; \boldsymbol{\theta}\right) = -\log p_{\boldsymbol{\theta}}\left(Y = y|\boldsymbol{x}\right) \tag{1}$$

where $p_{\boldsymbol{\theta}}$ is the conditional probability distribution defined by the neural network whose trainable bias and weight parameters are collected in $\boldsymbol{\theta}$. An important note is that in this paper this probability distribution is not always defined over all output nodes of the network, but only over the "active nodes". This means that the normalization performed by the final softmax layer only takes into account these active nodes, and that learning is thus restricted to those nodes. For experiments performed according to the incremental task learning scenario, for which we use a "multihead" softmax layer, always only the nodes of the task under consideration are active. Typically this is the current task, but for replayed data it is the task that is (intended to be) replayed. For the incremental domain learning scenario always all nodes are active. For the incremental class learning scenario, the nodes of all tasks seen so far are active, both when training on current and on replayed data.

For the method DGR, there are also some subtle differences between the continual learning scenarios when generating hard targets for the inputs to be replayed. With the incremental task learning scenario, only the classes of the task that is intended to be replayed can be predicted (in each iteration the available replays are equally divided over the previous tasks). With the incremental domain learning scenario always all classes can be predicted. With the incremental class learning scenario only classes from up to the previous task can be predicted.

### A.1.2  DISTILLATION

The methods LwF, DGR+distill and IGR use distillation loss for their replayed data. For this, each input $\boldsymbol{x}$ to be replayed is labeled with a "soft target", which is a vector containing a probability for each active class. This target probability vector is obtained using a copy of the main model stored after finishing training on the most recent task, and the training objective is to match the probabilities predicted by the model being trained to these target probabilities (by minimizing the cross entropy between them). Moreover, as is common for distillation, these two probability distributions that we want to match are made softer by temporary raising the temperature $T$ of their models' softmax layers.[1] This means that before the softmax normalization is performed on the logits, these logits are first divided by $T$. For an input $\boldsymbol{x}$ to be replayed during training of task $K$, the soft targets are given by the vector $\tilde{\boldsymbol{y}}$ whose $i^{\text{th}}$ element is given by:

$$\tilde{y}_i = p_{\hat{\boldsymbol{\theta}}^{(K-1)}}^T\left(Y = i|\boldsymbol{x}\right) \tag{2}$$

where $\hat{\boldsymbol{\theta}}^{(K-1)}$ is the vector with parameter values at the end of training of task $K-1$ and $p_{\boldsymbol{\theta}}^T$ is the conditional probability distribution defined by the neural network with parameters $\boldsymbol{\theta}$ and with the temperature of its softmax layer raised to $T$. The distillation loss function for an input $\boldsymbol{x}$ labeled with a soft target vector $\tilde{\boldsymbol{y}}$ is then given by:

$$\mathcal{L}_{\text{distillation}}\left(\boldsymbol{x}, \tilde{\boldsymbol{y}}; \boldsymbol{\theta}\right) = -T^2 \sum_{i=1}^{N_{\text{classes}}} \tilde{y}_i \log p_{\boldsymbol{\theta}}^T\left(Y = i|\boldsymbol{x}\right) \tag{3}$$

where the scaling by $T^2$ is included to ensure that the relative contribution of this objective matches that of a comparable objective with hard targets Hinton et al. (2015).

---

[1] The same temperature should be used for calculating the target probabilities and for calculating the probabilities to be matched during training; but during testing the temperature should be set back to 1. A typical value for this temperature is 2, which is the value used in this paper.

When generating soft targets for the inputs to be replayed, there are again subtle differences between the three continual learning scenarios. With the incremental task learning scenario, the soft target probability distribution is defined only over the classes of the task intended to be replayed. With the incremental domain learning scenario this distribution is always over all classes. With the incremental class learning scenario, the soft target probability distribution is first generated only over the classes from up to the previous task and then zero probabilities are added for all classes in the current task.

## A.1.3 SYMMETRICAL VAE

The separate generative model that is used for DGR and DGR+distill is a variational autoencoder (VAE; Kingma & Welling, 2013), of which both the encoder network $q_\phi$ and the decoder network $p_\psi$ are multi-layer perceptrons with 2 hidden layers containing 400 (split MNIST) or 1000 (permuted MNIST) units with ReLU non-linearity. The stochastic latent variable layer $z$ has 100 units and the prior over them is the standard normal distribution. Following Kingma & Welling (2013), the "latent variable regularization term" of this VAE is given by:

$$\mathcal{L}_{\text{latent}}(\boldsymbol{x}, \boldsymbol{\phi}) = \frac{1}{2} \sum_{j=1}^{100} \left( 1 + \log\left( \left( \sigma_j^{(\boldsymbol{x})} \right)^2 \right) - \left( \mu_j^{(\boldsymbol{x})} \right)^2 - \left( \sigma_j^{(\boldsymbol{x})} \right)^2 \right) \tag{4}$$

whereby $\mu_j^{(\boldsymbol{x})}$ and $\sigma_j^{(\boldsymbol{x})}$ are the $j^{\text{th}}$ elements of respectively $\boldsymbol{\mu}^{(\boldsymbol{x})}$ and $\boldsymbol{\sigma}^{(\boldsymbol{x})}$, which are the outputs of the encoder network $q_\phi$ given input $\boldsymbol{x}$. Following Doersch (2016), the output layer of the decoder network $p_\psi$ has a sigmoid non-linearity and the "reconstruction term" is given by the binary cross entropy between the original and decoded pixel values:

$$\mathcal{L}_{\text{recon}}(\boldsymbol{x}; \boldsymbol{\phi}, \boldsymbol{\psi}) = \sum_{i=1}^{N_{\text{pixels}}} x_i \log\left(\tilde{x}_i\right) + (1 - x_i) \log\left(1 - \tilde{x}_i\right) \tag{5}$$

whereby $x_i$ is the value of the $i^{\text{th}}$ pixel of the original input image $\boldsymbol{x}$ and $\tilde{x}_i$ is the value of the $i^{\text{th}}$ pixel of the decoded image $\tilde{\boldsymbol{x}} = p_\psi\left(\boldsymbol{z}^{(\boldsymbol{x})}\right)$ with $\boldsymbol{z}^{(\boldsymbol{x})} = \boldsymbol{\mu}^{(\boldsymbol{x})} + \boldsymbol{\sigma}^{(\boldsymbol{x})} \cdot \boldsymbol{\epsilon}$, whereby $\boldsymbol{\epsilon}$ is sampled from $\mathcal{N}\left(0, \boldsymbol{I}_{100}\right)$. The per-sample VAE loss for an input $\boldsymbol{x}$ is then given by (Kingma & Welling, 2013):

$$\mathcal{L}_{\text{generative}}(\boldsymbol{x}; \boldsymbol{\phi}, \boldsymbol{\psi}) = \mathcal{L}_{\text{recon}}(\boldsymbol{x}; \boldsymbol{\phi}, \boldsymbol{\psi}) + \mathcal{L}_{\text{latent}}(\boldsymbol{x}; \boldsymbol{\phi}) \tag{6}$$

The two terms of this loss function correspond to the two objectives mentioned in the main text: $\mathcal{L}_{\text{recon}}$ encourages the backward connections to be able to reconstruct the inputs from their hidden representations and $\mathcal{L}_{\text{latent}}$ encourages—given the observed inputs—the distribution of the latent variables to be close to a standard normal distribution.

The model used for IGR is equal to the symmetrical VAE described above, except for an added softmax classification layer to the last hidden layer of the encoder. As explained in the main text, the per-sample loss of this model is simply the sum of a classification term (either standard cross-entropy loss or distillation loss) and the above generative loss term. It would be possible to add a hyperparameter to set the relative weight of these two terms (which would presumably further increase performance), but given the issue associated with hyperparameters in a continual learning setting (see Appendix B), we choose to avoid this.

## A.2 REGULARIZATION TERMS

### A.2.1 EWC

The regularization term of elastic weight consolidation (EWC; Kirkpatrick et al., 2017) consists of a quadratic penalty term for each previously learned task, whereby each task's term penalizes the parameters for how different they are compared to their value directly after finishing training on that task. The strength of each parameter's penalty depends for every task on how important that parameter was estimated to be for that task, with higher penalties for more important parameters. For EWC, a parameter's importance is estimated for each task by the parameter's corresponding diagonal element of that task's Fisher Information matrix, evaluated at the optimal parameter values after finishing training on that task. The EWC regularization term for task $K > 1$ is given by:

$$\mathcal{L}_{\text{regularization}_{\text{EWC}}}^{(K)}(\boldsymbol{\theta}) = \sum_{k=1}^{K-1} \left( \frac{1}{2} \sum_{i=1}^{N_{\text{params}}} F_{ii}^{(k)} \left( \theta_i - \hat{\theta}_i^{(k)} \right)^2 \right) \tag{7}$$

whereby $\hat{\theta}_i^{(k)}$ is the $i^{\text{th}}$ element of $\hat{\theta}^{(k)}$, which is the vector with parameter values at the end of training of task $k$, and $F_{ii}^{(k)}$ is the $i^{\text{th}}$ diagonal element of $\boldsymbol{F}^{(k)}$, which is the Fisher Information matrix of task $k$ evaluated at $\hat{\theta}^{(k)}$. Following the definitions and notation in Martens (2014), the $i^{\text{th}}$ diagonal element of $\boldsymbol{F}^{(k)}$ is defined as:

$$F_{ii}^{(k)} = \mathbb{E}_{\boldsymbol{x} \sim Q_{\boldsymbol{x}}^{(k)}} \left[ \mathbb{E}_{p_{\boldsymbol{\theta}}(y|\boldsymbol{x})} \left[ \left( \frac{\delta \log p_{\boldsymbol{\theta}}\left(Y = y|\boldsymbol{x}\right)}{\delta \theta_i} \right)^2 \right] \right] \Bigg|_{\boldsymbol{\theta} = \hat{\boldsymbol{\theta}}^{(k)}} \tag{8}$$

whereby $Q_{\boldsymbol{x}}^{(k)}$ is the (theoretical) input distribution of task $k$ and $p_{\boldsymbol{\theta}}$ is the conditional distribution defined by the neural network with parameters $\boldsymbol{\theta}$. Note that in Kirkpatrick et al. (2017) it is not specified exactly how these $F_{ii}^{(k)}$ are calculated (except that it is said to be "easy"); here we calculate them as the diagonal elements of the "empirical Fisher" (Martens, 2014):

$$F_{ii}^{(k)} = \frac{1}{|S^{(k)}|} \sum_{(\boldsymbol{x}, y) \in S^{(k)}} \left( \frac{\delta \log p_{\boldsymbol{\theta}}\left(Y = y|\boldsymbol{x}\right)}{\delta \theta_i} \bigg|_{\boldsymbol{\theta} = \hat{\boldsymbol{\theta}}^{(k)}} \right)^2 \tag{9}$$

whereby $S^{(k)}$ is the training data of task $k$. This calculation can be interpreted as an approximation under the assumption that the predictions made by $p_{\hat{\theta}^{(k)}}$ on the training data of task $k$ are near-perfect.[2] The calculation of the Fisher Information is time-consuming, especially if tasks have a lot of training data.[3] In practice it might therefore sometimes be beneficial to trade accuracy for speed by using only a subset of a task's training data for this calculation (e.g., by introducing another hyperparameter $N_{\text{Fisher}}$ that sets the maximum number of samples to be used in equation 9).

### A.2.2 ONLINE EWC

A disadvantage of the original formulation of EWC is that the number of quadratic terms in its regularization term grows linearly with the number of tasks. This is an important limitation, as for a method to be applicable in a true lifelong learning setting its computational cost should not increase with the number of tasks seen so far. It was pointed out by Huszár (2018) that a slightly stricter adherence to the approximate Bayesian treatment of continual learning, which had been used as motivation for EWC, actually results in only a single quadratic penalty term on the parameters that is anchored at the optimal parameters after the most recent task and with the weight of the parameters' penalties determined by a running sum of the previous tasks' Fisher Information matrices. This insight was adopted by Schwarz et al. (2018), who proposed a modification to EWC called *online EWC*. The regularization term of online EWC when training on task $K > 1$ is given by:

$$\mathcal{L}_{\text{regularization}_{\text{oEWC}}}^{(K)} = \sum_{i=1}^{N_{\text{params}}} \tilde{F}_{ii}^{(K-1)} \left( \theta_i - \hat{\theta}_i^{(K-1)} \right)^2 \tag{10}$$

whereby $\hat{\theta}_i^{(K-1)}$ is the value of parameter $i$ after finishing training on task $K - 1$ and $\tilde{F}_{ii}^{(K-1)}$ is a running sum of the $i^{\text{th}}$ diagonal elements of the Fisher Information matrices of the first $K - 1$ tasks, with a hyperparameter $\gamma \leq 1$ that governs a gradual decay of each previous task's contribution. That is: $\tilde{F}_{ii}^{(k)} = \gamma \tilde{F}_{ii}^{(k-1)} + F_{ii}^{(k)}$, with $\tilde{F}_{ii}^{(1)} = F_{ii}^{(1)}$ and $F_{ii}^{(k)}$ is the $i^{\text{th}}$ diagonal element of the Fisher Information matrix of task $k$ calculated according to equation 9.

### A.2.3 SI

Similar as for online EWC, the regularization term of synaptic intelligence (SI; Zenke et al., 2017) consists of only one quadratic term that penalizes changes to parameters away from their values after

---

[2] An alternative way to calculate $F_{ii}^{(k)}$ is to replace in equation 9 the provided label $y$ by $\hat{y}_{\boldsymbol{x}}^{(k)} = \arg\min_y \log p_{\hat{\theta}^{(k)}}\left(Y = y|\boldsymbol{x}\right)$, the label predicted by the model with parameters $\hat{\theta}^{(k)}$ given $\boldsymbol{x}$. Another option is, instead of taking for each training input $\boldsymbol{x}$ only the most likely label predicted by model $p_{\hat{\theta}^{(k)}}$, to sample for each $\boldsymbol{x}$ multiple labels from the entire conditional distribution defined by this model (i.e., to approximate the inner expectation of equation 8 for each training sample $\boldsymbol{x}$ with Monte Carlo sampling from $p_{\hat{\theta}^{(k)}}\left(\cdot|\boldsymbol{x}\right)$).

[3] It should be noted that it might be possible to improve the efficiency of our implementation of the Fisher Information calculation. This calculation requires the gradients for each individual data point (as they need to be squared before being summed), but batch-wise operations in PyTorch do not allow access to the unaggregated gradients. We therefore performed the Fisher Information calculation with a batch size of 1.

finishing training on the previous task, with the strength of each parameter's penalty depending on how important that parameter is thought to be for the tasks learned so far. To estimate parameters' importance, for every new task $k$ a per-parameter contribution to the change of the loss is first calculated for each parameter $i$ as follows:

$$\omega_i^{(k)} = \sum_{t=1}^{N_{\text{iters}}} \left( \theta_i[t^{(k)}] - \theta_i[(t-1)^{(k)}] \right) \frac{-\delta\mathcal{L}_{\text{total}}[t^{(k)}]}{\delta\theta_i} \tag{11}$$

with $N_{\text{iters}}$ the total number of iterations per task, $\theta_i[t^{(k)}]$ the value of the $i^{\text{th}}$ parameter after the $t^{\text{th}}$ training iteration on task $k$ and $\frac{\delta\mathcal{L}_{\text{total}}[t^{(k)}]}{\delta\theta_i}$ the gradient of the loss with respect to the $i^{\text{th}}$ parameter during the $t^{\text{th}}$ training iteration on task $k$. For every task, these per-parameter contributions are normalized by the square of the total change of that parameter during training on that task plus a small dampening term $\xi$ (set to 0.1, to bound the resulting normalized contributions when a parameter's total change goes to zero), after which they are summed over all tasks so far. The estimated importance of parameter $i$ for the first $K - 1$ tasks is thus given by:

$$\Omega_i^{(K-1)} = \sum_{k=1}^{K-1} \frac{\omega_i^{(k)}}{\left(\Delta_i^{(k)}\right)^2 + \xi} \tag{12}$$

with $\Delta_i^{(k)} = \theta_i[N_{\text{iters}}^{(k)}] - \theta_i[0^{(k)}]$, where $\theta_i[0^{(k)}]$ indicates the value of parameter $i$ right before starting training on task $k$. (An alternative formulation is $\Delta_i^{(k)} = \hat{\theta}_i^{(k)} - \hat{\theta}_i^{(k-1)}$, with $\hat{\theta}_i^{(0)}$ the value of parameter $i$ it was initialized with and $\hat{\theta}_i^{(k)}$ its value after finishing training on task $k$.) The regularization term of SI to be used during training on task $K$ is then given by:

$$\mathcal{L}_{\text{regularization}_{\text{SI}}}^{(K)} = \sum_{i=1}^{N_{\text{params}}} \Omega_i^{(K-1)} \left( \theta_i - \hat{\theta}_i^{(K-1)} \right)^2 \tag{13}$$

## B  HYPERPARAMETERS

As discussed in section 4.2 and Appendix A.2, several of the in this paper compared continual learning methods have one or more hyperparameters. The typical way of setting the value of hyperparameters is by training models on the training set for a range of hyperparameter-values, and selecting those that result in the best performance on a separate validation set. This strategy has been adapted to the continual learning setting as training models on the full protocol with different hyperparameter-values using only every task's training data, and comparing their overall performances using separate validation sets (or sometimes the test sets) for each task (e.g., see Goodfellow et al., 2013; Kirkpatrick et al., 2017; Kemker et al., 2017; Schwarz et al., 2018). However, here we would like to stress that this means that these hyperparameters are set (or learned) based on an evaluation using data from all tasks, which violates the continual learning principle of only being allowed to visit each task once and in sequence. Although it is tempting to think that it is acceptable to relax this principle for tasks' validation data, we argue here that it is not. A clear example of how using each task's validation data continuously throughout an incremental training protocol can lead to an in our opinion unfair advantage is provided by Wu et al. (2018), in which after finishing training on each task a "bias-removal parameter" is set that optimizes performance on the validation sets of all tasks seen so far (see their section 3.3). Although the hyperparameters of the methods compared here are much less influential than those in the above paper, we believe that it is important to realize this issue associated with traditional grid searches in a continual learning setting and that at a minimum influential hyperparameters should be avoided in methods for continual learning.

Nevertheless, to give the competing methods of generative replay the best possible chance—and to explore how influential their hyperparameters are—we do perform grid searches to set the values of their hyperparameters (see Figures 7 and 8). Given the issue discussed above we do not see much value in using validation sets for this, and we evaluate the performances of all hyperparameter(-combination)s using the tasks' test sets. For this grid search each experiment is run once, after which 20 new runs are executed using the selected hyperparameter-values to obtain the results in Tables 3 and 4 and Figures 4 and 5.

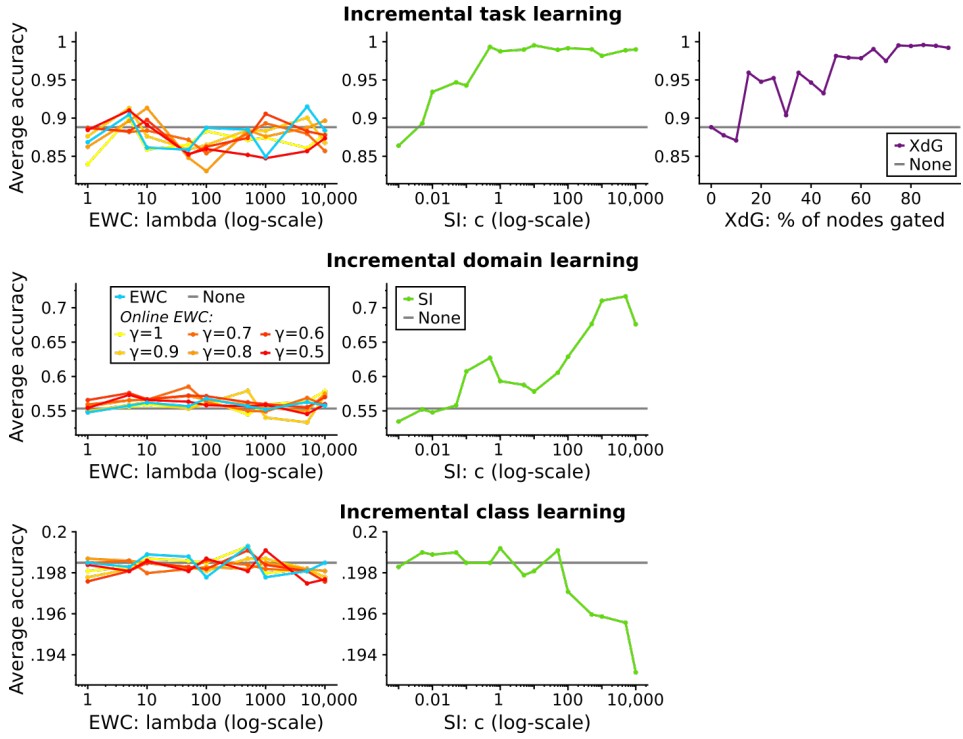

Figure 7: Grid searches for the split MNIST task protocol. Shown are the average test set accuracies (over all 5 tasks) for the (combination of) hyperparameter-values tested for each method.

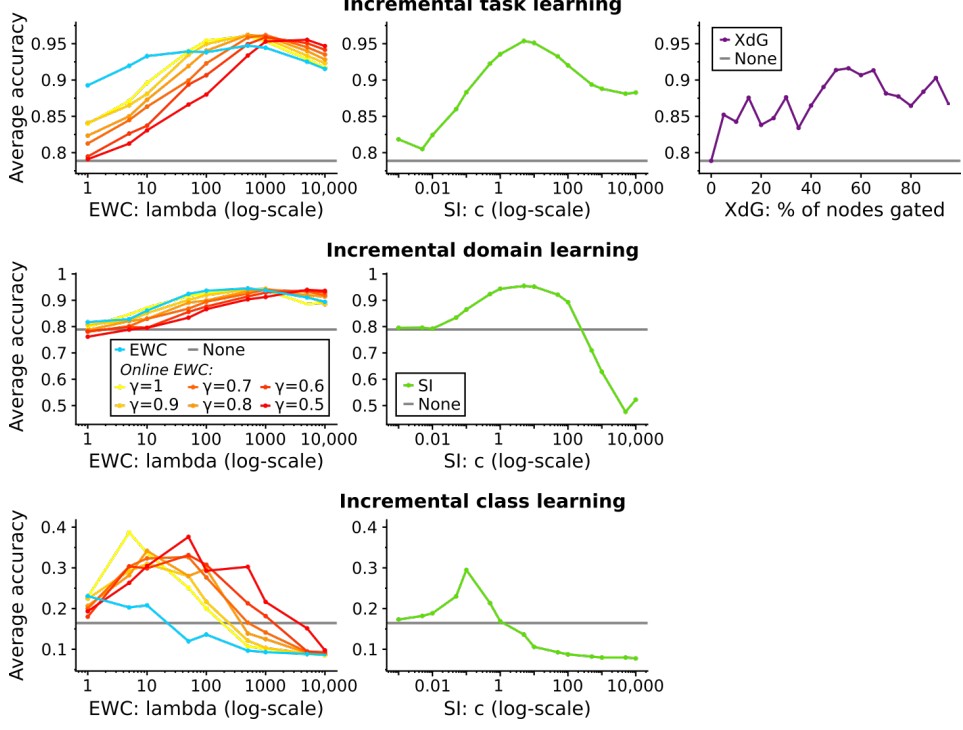

Figure 8: Grid searches for the permuted MNIST task protocol. Shown are the average test set accuracies (over all 10 tasks) for the (combination of) hyperparameter-values tested for each method.

## C  ADDITIONAL DISCUSSION: SIMILAR METHODS & STORING DATA

The Integrated Generative Replay framework we introduce here has some similarity with one of the components of FearNet (Kemker & Kanan, 2018). The "medial prefrontal cortex (mPFC) network" of this brain-inspired method is also a generative autoencoder with added classification capability. At specific times (whenever their model "sleeps"), this autoencoder is also used to generate pseudo-examples representative of previously learned tasks. This autoencoder is however not a VAE, but its generative capability relies on to the storage of a mean feature vector and a covariance matrix for each encountered class. FearNet further consists of a "hippocampal complex (HC) network", which temporary stores the data from previous tasks[4], and a "basolateral amygdala (BLA) network" that decides whether a newly-encountered example should be classified by the HC or the mPFC. Kemker & Kanan (2018) show good results with this method on a by them introduced variant of the incremental class learning scenario: models are first non-incrementally trained on half of all classes to be learned, after which in the incremental learning part of their paradigm the remaining classes are presented one-by-one (i.e., each new task only consists of one class). As this experimental paradigm is quite specific, it remained unclear how generally applicable their approach is. Moreover, due to the complexity of their method, the specific contribution of replay generated by their autoencoder remained unclear. Indeed, it seems likely that especially the storage of data—the temporary storage of original data and/or the permanent storage of hidden summary statistics—also had an important role in FearNet's good performance.

In the current paper, based on the argument that storing data is not always possible due to privacy concerns or memory constraints, we only considered methods that do not store data. However, as indicated by methods such as iCaRL (Rebuffi et al., 2017) and FearNet (Kemker & Kanan, 2018), when possible, storing data can substantially boost performance, especially in the incremental class learning scenario. Of particular note is that these two methods point out that it is not necessary that all of the original data is stored permanently. Indeed, iCaRL demonstrates that storing a relatively small number of well-chosen examples can be helpful, while—as discussed above—FearNet's good performance seems to suggest that temporary storage of data can already be useful. Both these reductionist approaches to storing data of course reduce memory storage demands. Finally, an interesting aspect of FearNet is that it also stored hidden summary statistics. The promising idea of storing hidden representations, which besides reducing memory storage demands could also address the privacy issue, is further worked out in Riemer et al. (2018). We expect that the sparse and/or temporary storage of hidden representations could be a useful complement to generative replay, which might help it to scale up to real-world continual learning problems. However, we want to stress that when allowing the storage of data, it is important to take extra care to ensure fair comparisons between methods (i.e., that they all have the same rules regarding to how much, for how long and what data can be stored).

---

[4]The data from previous tasks is stored in the HC until their model "sleeps", which for their main reported results is every ten tasks. Confusingly, in the abstract of Kemker & Kanan (2018) it is claimed that their method does not store previous examples; this was probably intended as that their method does not *permanently* store previous examples.

