# OpenReview forum: "Three continual learning scenarios and a case for generative replay"
_ICLR.cc/2019/Conference_

### Official Review · AnonReviewer3 · 2018-11-01
**The comparison study needs some more detail, the RtF part is not significant enough.**

**Rating:** 6
**Confidence:** 5

**Review:**

This paper points out a important issue in current continual learning literature: Due to the different settings and different evaluation protocols of each method, comparison between methods are usually not fair, and lead to distinct conclusions.
The paper is in general easy to understand except a few drawbacks listed in the cons.

Pros:
1. This paper investigates an important problem, aka, how does the methods compare to each other with the same evaluation protocol.
2. Experiments are performed on the previous methods, which could be used as a baseline for future works in this field.
3. Proposes to combine discriminative model with generative model to save computation when using generative model to store rehearsal examples.

Cons:
1. Details of each experiments are missing.
Different methods are evaluated under the "incremental task learning", "incremental domain learning",  "incremental class learning" settings. However, to my knowledge, some of the methods will not work under all of the three settings, as the author also suggest that XdG only works with task id. However, I think there are a few more. For example, the LwF methods has multiple sets of output neurons, which implicitly assumes the task id is known. It is not described in the paper how to evaluate it under "incremental domain learning", aka, how to decide which set of output to use if task id is not available during testing. Another example, the results in table 3 and 4 indicates that EWC with task id is better than without. However, original EWC does not take task id during testing, it is not described how to introduce dependency on the task id for EWC.
2. Using the term feedback connection is misleading to the reader since the described method is just using an encoder/decoder structure. In my opinion this is different from feedback connection in which higher layer is an input for lower layers. Autoencoder or encoder/decoder structure is more appropriate.
3. There is some contribution in the RtF part, namely the saved computation compared to DGR. However, subjectively, I think this contribution is not very significant. The same thing can be achieved with DGR by sharing the network between the discriminative model and the discriminator in GAN. In my opinion this is more a design bonus in using generative replay than a major methodology innovation.

Conclusion:
The first part that compares different methods is worth publishing given more details are provided. I'm more than happy to give a higher score if the authors are able to provide more details and the details are reasonable.

---

> ### Author Response · Authors · 2018-11-24
> **Response to reviewer 3**
>
> We thank the reviewer for his/her review and for the positive feedback on our proposed evaluation framework. Please find our responses to the three “Cons” below.
>
> 1) Firstly, the reviewer is correct in that some important experimental details were missing from the main text. We sincerely apologize for this and thank the reviewer for pointing this out. We have now included these missing details towards the top of section 4.2.
> As the reviewer points out, the original LwF method was implemented with a “multihead” output layer with a different set of output units for each task. This implementation indeed requires task identity during testing and can thus only be used in the incremental task learning scenario. To be able to use (the core elements of) LwF (i.e., replaying current task inputs labelled with soft targets produced by the previous model) in the incremental domain learning scenario, for that scenario we instead use a “singlehead” output layer, whereby all tasks share the same set of output units.
> Conversely, the original EWC method was implemented with a “singlehead” output layer. This implementation of EWC indeed does not use task identity during testing. As we have now described in our methods-section, to enable EWC to use the available task identity information in the incremental task learning scenario, for that scenario we use a “mutlihead” output layer instead.
> (Finally, note that in the incremental class learning scenario, all methods use an output layer with a separate output unit for each class and always all units of the classes seen so far are “active”.)
>
> 2) Our use of the term “feedback connections” was motivated by Neuroscience, where connections from a higher processing region back to a lower processing region are referred to as feedback connections. We believe this parallel to be appropriate (to some degree) for the encoder/decoder structure that we used. But we do agree with the reviewer that there are other uses of the term “feedback” that do not align with our proposed method and that it could indeed be misinterpreted. We have therefore now removed all mentions of “feedback” from our paper and we changed the name “Replay-through-Feedback (RtF)” to “Integrated Generative Replay (IGR)”. We have also made an effort to make sure it is clear that we use an encoder/decoder structure.
>
> 3) We agree with the reviewer that it would also be possible to integrate the classifier into a GAN by letting it share a network with the discriminator, which would indeed save computation compared to DGR implemented with a GAN. We believe this to be along the same spirit as what we propose to do with a VAE. We had considered this option as well, but we expect that, due to the generally higher computational costs of training a GAN compared to a VAE, overall this would be more computationally expensive than our implementation. We also expect that it is more beneficial / less harmful to share weights between the classifier and a VAE-encoder than between the classifier and a GAN-discriminator, because we expect the representations extracted by a VAE-encoder (from which it should be possible to reconstruct the full input) to be more useful for robust classification than the representations extracted by a GAN-discriminator (from which it only needs to be possible to make a distinction between real and generated inputs). But these are merely expectations and a formal comparison between these two approaches would be interesting.

---

> > ### Comment · AnonReviewer3 · 2018-12-09
> > **Keep the score**
> >
> > I feel the rebuttal is not sufficient for me to increase the score.
> > I would suggest a rewrite of the paper to separate the two topics in future submission.
> > Experimental study of the methods under different evaluation metrics deserve more length.

---

### Official Review · AnonReviewer2 · 2018-11-03
**VAE with additional softmax classification layer for continual learning**

**Rating:** 4
**Confidence:** 4

**Review:**

summary: The paper claims to make three contributions
1. It surveys the current literature on preventing catastrophic forgetting during lifelong learning. It explains the apparent inconsistencies in reported results by distinguishing three types of deployment scenarios, categorizing the evaluation procedures in the literature accordingly.
2. The paper conducts two sets of simulated experiments on MNIST data to understand which existing methods (do not) work well. It finds that deep generative replay (DGR) that learns to generate imaginary new samples from previously seen training data, potentially augmented with soft labels seems to work best in these specific experiments but potentially doubles the computational cost.
3. To reduce computational cost without sacrificing much accuracy it proposes to integrate the ability to learn to generate imaginary samples into the learning of the classifier itself. It does this by augmenting a symmetrical VAE with a softmax classification layer connected to the final hidden layer of the encoder.

Comments about significance:
1. I'm not entirely sure if the paper does a good job separating contributions 2 & 3 above cleanly so that each can stand on its own and be fully trust-worthy.
2. In particular, the experimental evaluation depends on the NN architectures chosen. Here the choice of architectures that were used for the best performing approach in the experiment (DGR & the classifier) were simply combined together to motivate the new approach. However, this feels a bit too simplistic. for example, what would happen if you replaced the simple 2-hidden-layer NN with a much more sophisticated network for each classifier, but still had a simple VAE to generate samples? the combination is no longer likely to be this easy but it would likely work more accurately than anything shown in table 3.

Novelty: This reviewer feels that augmenting a 2-hidden-layer VAE with a softmax classification layer does not seem to be a very significant new contribution by itself. The fact that it is being motivated for the specific problem of reducing catastrophic forgetting during lifelong learning is the main novelty here, but the relative amount of novelty might to be somewhat limited when viewed from this perspective.

---

> ### Author Response · Authors · 2018-11-24
> **Response to reviewer 2**
>
> We thank the reviewer for his/her review. Note however that we are not completely sure whether we fully understood the specific concern raised by this reviewer; we would appreciate a correction if we missed or misinterpreted something.
>
> As we understand it, the main concern raised by this reviewer is that the reduction in computational cost achieved by our RtF method is dependent on the NN architectures used. Specifically, the reviewer gives as example that it would be more difficult to combine a sophisticated classifier network with a simple generator network, while the reviewer thinks that this combination would be more accurate than any of our reported results.
> We agree that our proposed framework likely yields most benefit when the sophistication (depth) of the classifier and generator are reasonably balanced, but we don’t think that in practice this is an important limitation to the ML practitioner. Because although enhancing the sophistication of the classifier could increase the overall accuracy (by increasing the baseline accuracy on each individual task), so could enhancing the sophistication of the generator (by improving the quality of the replayed samples). For most practical applications, we expect the ‘optimal sophistication’ (in terms of accuracy vs. efficiency) of the classifier and generator to be reasonably biased.
>
> Regarding novelty, significance and broad impact of our work for the ML community, as discussed above we believe that the three continual learning scenarios we defined here are an important and much-needed contribution to the continual learning field that has already began to be adopted by the field [1]. In addition, our finding that currently only replay-based methods—and not ones using regularization-based approaches—are able to solve the most relevant catastrophic forgetting problem is novel and we believe of broad interest to the field. We have now rewritten the paper to better reflect these contributions, and we hope that the reviewer would be willing to reconsider her/his evaluation of our work.
>
> [1] Hsu, Liu & Kira (2018) “Re-evaluating Continual Learning Scenarios: A Categorization and Case for Strong Baselines.” arXiv preprint arXiv:1810.12488.

---

### Official Review · AnonReviewer1 · 2018-11-05
**An incremental method on lifelong learning.**

**Rating:** 4
**Confidence:** 4

**Review:**

This paper summarizes previous lifelong learning methods and identifies three different continual learning scenarios. Based on that, it draws a conclusion that DGR+distill outperforms other methods on all these scenarios. Further, the paper proposes unified model that combines a replay generator and a classification model. The proposed RTF model achieves comparable performance with DGR+distill and is approximately two times faster than DGR+distill.

My biggest concern is the novelty of the model, since RTF is still a replay-based method that is very similar as DGR+distill. Empirically it can be expected that RTF should behave similar as DGR+distill as well. And the result in this paper justifies that. So the main contribution comes from the efficiency boost by the integrated model strategy. That is, by replacing a separate generative model by a symmetric VAE. Besides that, there seem to be no significant contribution of the proposed model.

In my opinion, this paper look somewhat incremental. The first five pages are mostly reviews of previous methods, and the model it propose behave very similar to a previous method.

---

> ### Author Response · Authors · 2018-11-24
> **Response to reviewer 1**
>
> We thank the reviewer for his/her review.
> This reviewer’s main criticism is that our proposed RtF method is very similar to a previous method (i.e., DGR+distill; although note that this is actually a hybrid of two published methods) and that besides approximately doubling efficiency there seem to be no significant contributions of the proposed model. We agree that our demonstration that RTF and DGR+distill have comparable performance is not necessarily surprising. However, we do believe that doubling efficiency is an important contribution given that a critical feature determining the practical applicability of any continual learning method is its computational efficiency.
>
> Regarding novelty, significance and broad impact of our work for the ML community, as discussed above we believe that the three continual learning scenarios we defined here are an important and much-needed contribution to the continual learning field that has already began to be adopted by the field [1]. In addition, our finding that currently only replay-based methods—and not ones using regularization-based approaches—are able to solve the most relevant catastrophic forgetting problem is novel and we believe of broad interest to the field. We have now rewritten the paper to better reflect these contributions, and we hope that the reviewer would be willing to reconsider her/his evaluation of our work.
>
> [1] Hsu, Liu & Kira (2018) “Re-evaluating Continual Learning Scenarios: A Categorization and Case for Strong Baselines.” arXiv preprint arXiv:1810.12488.

---

> > ### Comment · AnonReviewer2 · 2018-12-08
> > **I still feel the same way about limited amount of novelty after the discussion**
> >
> > Thank you for your response. However I do not feel I can support the paper for the same reasons I mentioned in my original review.

---

### Author Response · Authors · 2018-11-24
**General response to all reviewers**

We thank all reviewers for their time and effort for assessing our study and their valuable comments. We address them point by point below. Based on their feedback we have made changes to the manuscript, which we believe have substantially improved it.

Their main criticism is that our proposed method of “Replay-through-Feedback” is not a novel enough technical advance to warrant acceptance at ICLR. Although we have a few specific comments on this point (please see below), in general we agree with this assessment. However, the major novel contribution of our paper was not meant to be the technical advance of the “Replay-through-Feedback” architecture but instead: a) we identify three distinct continual learning scenarios that provide a much-needed structured and systematic evaluation framework of the proposed solutions to catastrophic forgetting; and b) we find that the three continual learning scenarios we defined significantly differentiate the previously proposed solutions to catastrophic forgetting grouping proposed solutions into two major classes. First, methods that use regularization-based approaches (i.e., synaptic metaplasticity) can solve the easy continual learning scenarios, but exhibit catastrophic forgetting under the more ethological relevant scenario where an agent also needs to recognize which task needs to be performed. Importantly, it is under this scenario that humans and animals typically have to continuously learn and execute in order to survive. Second, methods with generative replay can perform well under this more difficult scenario overcoming the problem of catastrophic forgetting. Given that solving catastrophic forgetting is a fundamental problem for the machine learning community in the quest to achieve human-level performance in lifelong learning, we believe our contribution is novel and of board interest. Our findings underscore the importance of evaluating proposed solutions under the different continual learning scenarios, which has not been widely used to-date (e.g., [1]), and they highlight that learning a generative model of the tasks seems to be a promising path forward for successfully creating lifelong learning AI systems.

Additionally, we also motivate and introduce a new way of evaluating continual learning methods: in terms of both their accuracy and their efficiency. In the lifelong learning setting, models need to be continually trained on new tasks, and in many practical applications this training even has to be done in real-time, for example in situations that can be unpredictable and where the ability to quickly adapt to dynamic circumstances is of primary importance. Efficiency is therefore very important for any continual learning method that is to be used in practice, and we believe it should be taken into account when comparing methods.

We acknowledge that the way we had written the paper did not make our novel contributions clear enough. We have now made major changes to better highlight our contributions and their significance. We hope that after these revisions, reviewers 1 and 2 are willing to reassess their evaluation of our study.

[1] Masse, Grant & Freedman (2018) “Alleviating catastrophic forgetting using context-dependent gating and synaptic stabilization.” PNAS, 115(44): E10467-E10475.

---

> ### Public Comment · (anonymous) · 2018-12-17
> **Clarification on the novelty**
>
> What you claim as the main/ novel contributions have already been discussed by [1] and [2], albeit with different terminologies. Can you please comment on how 'Incremental task learning' and 'Incremental class learning' is different from multi-head and single-head setting, respectively?
>
> [1] Chaudhry et al. Riemannian Walk for Incremental Learning: Understanding Forgetting and Intransigence, ECCV, 2018.
> [2] Farquhar, S., & Gal, Y. (2018). Towards Robust Evaluations of Continual Learning. Lifelong Learning: A Reinforcement Learning Approach Workshop at ICML 2018.

---

> > ### Author Response · Authors · 2018-12-19
> > **Three scenarios and not tied to specific architectural layout**
> >
> > Thank you for the comment and for pointing us to these two papers, both of which indeed also discuss the need for defining different learning scenarios to evaluate continual learning algorithms.
> >
> > Compared with these two studies, our treatment of this important problem is more general. The mentioned papers both focus on the split MNIST task protocol (or other split dataset protocols), for which they discuss that there is a difference between whether models are evaluated with a multi-head layout (which requires task identity to be known) or with a single-head layout (which does not require task identity to be known). In our work we go further in two important ways.
> > First, we note that when task-identity is not known, there is a further distinction depending on whether task-identity needs to be inferred by the model as well. The two mentioned papers do not discuss this, but we believe this to be important since two distinct scenarios arise, both with relevant real-world applications. When task identity does not need to be inferred, the resulting scenario is incrementally learning new domains. A real-world example is an agent who needs to learn to survive in different environments without the need to explicitly identify the environment it is confronted with. On the other hand, when task identity does need to be inferred, the resulting scenario corresponds to incrementally learning new classes. “Single-headed split MNIST” is an example of this scenario. Importantly, in our paper we show—for both the split MNIST and the permuted MNIST task protocol—that there are substantial differences in difficulty between these two ethologically relevant scenarios, which are not captured by the multi-head / single-head split.
> > Second, while the multi-head / single-head distinction is tied to the architectural layout of the output layer, our identified scenarios are more general and reflect what information is available / what is required of a network. In the continual learning literature a multi-head layout (i.e., a model with a separate output layer for each task) might be the most common way to use task identity information, it is not the only way. Similarly, although a single-head layout (i.e., a model that uses the same output-layer for every task) by itself does not require task identity to be known, it is still possible for the model to use task identity information in other ways (e.g., in its hidden layers, as in context-dependent gating).

---

### Meta-Review · Area_Chair1 · 2018-12-14

**Confidence:** 5
**Recommendation:** Reject

**Metareview:**

The authors have proposed 3 continual learning variants which are all based on MNIST and which vary in terms of whether task ids are given and what the classification task is, and they have proposed a method which incorporates a symmetric VAE for generative replay with a class discriminator. The proposed method does work well on the continual learning scenarios and the incorporation of the generative model with the classifier is more efficient than keeping them separate. The discussion of the different CL scenarios and of related work is nice to read. However, the authors imply that these scenarios cover the space of important CL variants, yet they do not consider many other settings, such as when tasks continually change rather than having sharp boundaries. The authors have also only focused on the catastrophic forgetting aspect of continual learning, without considering scenarios where, e.g., strong forward transfer (or backwards transfer) is very important. Regarding the proposed architecture that combines a VAE with a softmax classifier for efficiency, the reviewers all felt that this was not novel enough to recommend publication.